# Engineering of a compact, high-fidelity EbCas12a variant that can be packaged with its crRNA into an all-in-one AAV vector delivery system

**Hongjian Wang**[�]**, Jin Zhou**[�]**, Jun Lei**[�]**, Guosheng Mo, Yankang Wu, Huan Liu, Ziyan Pang, Mingkun Du, Zihao Zhou, Chonil Paek, Zaiqiao Sun, Yongshun Chen, Yan Wang**\*, **Peng Chen**\*, **Lei Yin**[ORCID]\*

State Key Laboratory of Virology, Hubei Key Laboratory of Cell Homeostasis, College of Life Sciences, Department of Clinical Oncology, Renmin Hospital of Wuhan University, Wuhan University, Wuhan, China

☉ These authors contributed equally to this work.
\* Wang.y@whu.edu.cn (YW); chen-peng@whu.edu.cn (PC); yinlei@whu.edu.cn (LY)

**Data Availability Statement:** The EbCas12a gene GenBank accession number: SAAR01000186; Taxonomy ID: 2184014, Genome assembly: ASM991633v1. Raw sequencing reads of GUIDE-

## Abstract

The CRISPR-associated endonuclease Cas12a has become a powerful genome-editing tool in biomedical research due to its ease of use and low off-targeting. However, the size of Cas12a severely limits clinical applications such as adeno-associated virus (AAV)-based gene therapy. Here, we characterized a novel compact Cas12a ortholog, termed EbCas12a, from the metagenome-assembled genome of a currently unclassified *Erysipelotrichia*. It has the PAM sequence of 5′-TTTV-3′ (V = A, G, C) and the smallest size of approximately 3.47 kb among the Cas12a orthologs reported so far. In addition, enhanced EbCas12a (enEbCas12a) was also designed to have comparable editing efficiency with higher specificity to AsCas12a and LbCas12a in mammalian cells at multiple target sites. Based on the compact enEbCas12a, an all-in-one AAV delivery system with crRNA for Cas12a was developed for both in vitro and in vivo applications. Overall, the novel smallest high-fidelity enEbCas12a, this first case of the all-in-one AAV delivery for Cas12a could greatly boost future gene therapy and scientific research.

## Introduction

CRISPR (clustered regularly interspaced short palindromic repeats) and CRISPR-associated proteins, a part of the adaptive immune system of bacteria and archaea, provide attractive genome-editing tools for therapeutic applications [1–13]. The type II CRISPR associated with the Cas9 endonuclease (CRISPR/Cas9) system and the type V CRISPR associated with the Cas12a endonuclease (CRISPR/Cas12a) system are popular for various scientific researches. Through RNA-guided, CRISPR/Cas9 or CRISPR/Cas12a first recognizes the specific PAM region, locates in the targeted genomic loci, and then induces double-stranded DNA breaks [14]. However, there are some differences in the DNA cleavage process between Cas9 and

seq are available at the Gene Expression Omnibus under accession GSE236647. The deep-sequencing data from this study are available at the NCBI Sequence Read Archive under the accession number PRJNA1077696.

**Funding:** This work was supported by the National Key R&D Program of China (2022YFA1303500 to LY), the National Natural Science Foundation of China (32101196 to PC, 32171210, 31870728 and 32371271 to LY), the Fundamental Research Funds for the Central Universities (2042022kf1189 to LY), the China Postdoctoral Science Foundation (2021TQ0253 and 2022M712468 to PC, 2022M722473 to JZ). The funders had no role in study design, data collection and analysis, decision to publish, or preparation of the manuscript.

**Competing interests:** The authors have declared that no competing interests exist.

**Abbreviations:** AAV, adeno-associated virus; CRISPR, clustered regularly interspaced short palindromic repeats; crRNA, CRISPR RNA; DMEM, Dulbecco's modified Eagle's medium; DSB, double-stranded break; dsODN, double-stranded oligodeoxynucleotide; GC, genomic copy; LB, Luria–Bertani; PBS, phosphate-buffered saline; ssDNA, single-stranded DNA; tracRNA, transactivating crRNA.

Cas12a [15–19]. <1> Guide RNA of Cas9 is composed of a CRISPR RNA (crRNA) that contains complementary sequences to the target DNA and a transactivating crRNA (tracRNA) that contributes to the connection between Cas9 and crRNA. Cas12a only needs a single crRNA as guide RNA to direct DNA cleavage. <2> Cas9 recognizes a G-rich PAM followed by a target DNA spacer, while Cas12a recognizes a T-rich PAM which is on the 5′ side of the spacer. <3> Cas9 cleaves DNA by 2 nuclease domains (RuvC and HNH domains), generating a blunt-ended DSB near the PAM, while Cas12a uses a RuvC domain to make DSB and create 5′-overhang sticky ends at the far end of the PAM.

It is critical to deliver CRISPR/Cas systems to target cells for gene editing. Currently, many common methods are used for delivery, such as electroporation, nanoparticles, and viral vectors, depending on the types of cells or diseases that need gene editing [20]. Adeno-associated virus (AAV) is a single-stranded DNA virus, one of the most promising vectors that have been widely used for gene therapy. AAV has been employed safely and successfully in clinical trials [21,22]. Moreover, it is suitable and efficient for multiplex cells and tissue types with low immunogenicity and low associated toxicity [21,22]. In addition, unlike plasmid DNA and ribonucleoproteins that may be degraded in vivo after a few days, AAV delivery can provide an indefinite persistence of the provided DNA [21,22]. However, AAV has a small packaging capacity of <4.7 kb that limits clinical applications of most CRISPR/Cas system tools [23]. Although CjCas9, SaCas9, and SpaCas9 (some smaller Cas9) have been used as genome-editing tools with the sgRNA through a single AAV vector [13,24,25], Cas12a is too large to be encapsulated with crRNA. To adopt Cas12a in AAV gene therapy, the small Cas12as that can be packaged with its crRNA into a single AAV are urgently needed.

In this study, we characterize a novel small V-A Cas12a enzyme, termed EbCas12a, with a 5′-TTTV (V = A, G, C)-3′ PAM from the metagenome-assembled genome of a currently unclassified *Erysipelotrichia* [26]. By introducing a point mutation, D141R, we developed a high-activity variant, hereafter referred to as enhanced EbCas12a (enEbCas12a). enEbCas12a shows comparable efficiency to AsCa12a and LbCa12a, and approximately 1.9-fold that of wild-type EbCas12a. Furthermore, we found that enEbCas12a also exhibited low off-targeting through the genome-wide unbiased identification of double-stranded breaks (DSBs) enabled by sequencing (GUIDE-seq). Importantly, the all-in-one AAV delivery of enEbCas12a with its crRNA works well both in vitro and in vivo. Taken together, enEbCas12a provides a promising genome-editing tool and a powerful platform for AAV gene therapy.

## Result

### EbCas12a possesses DNA cleavage activities in vitro

We characterize a compact Cas12a ortholog from the metagenome-assembled genome of a currently unclassified *Erysipelotrichia* with the smallest size (1158a) among other Cas12a nucleases (**Fig 1A and 1B**). To further investigate whether the EbCas12a could cleave dsDNA, we purified the protein of EbCas12a-crRNA complex and incubated RNP with dsDNA substrate containing TTTA PAM at 37°C for different time points (0 min, 2.5 min, 5 min, 8 min, 10 min, and 15 min). The result showed that EbCas12a strongly cleaves dsDNA in vitro (**Fig 1C**). Next, the PAM depletion assay was conducted to determine the requirement of dsDNA targeted by EbCas12a. A linear dsDNA library containing 4 bp random PAM sequences was synthesized and then incubated with EbCas12a-crRNA RNP. Through NGS, we analyzed preferred and non-preferred PAMs for EbCas12a and found that EbCas12a also recognized 5′-TTTV (V = A, G, C) PAMs (**Figs 1D, 1E, and S1A**). A PAM preference profile for EbCas12a was shown by normalized maximum cleavage rate at 5′-TTTV PAM (**Fig 1D and 1E**). Although EbCas12a specifically recognized PAMs as 5′-TTTV, C-containing PAMs were

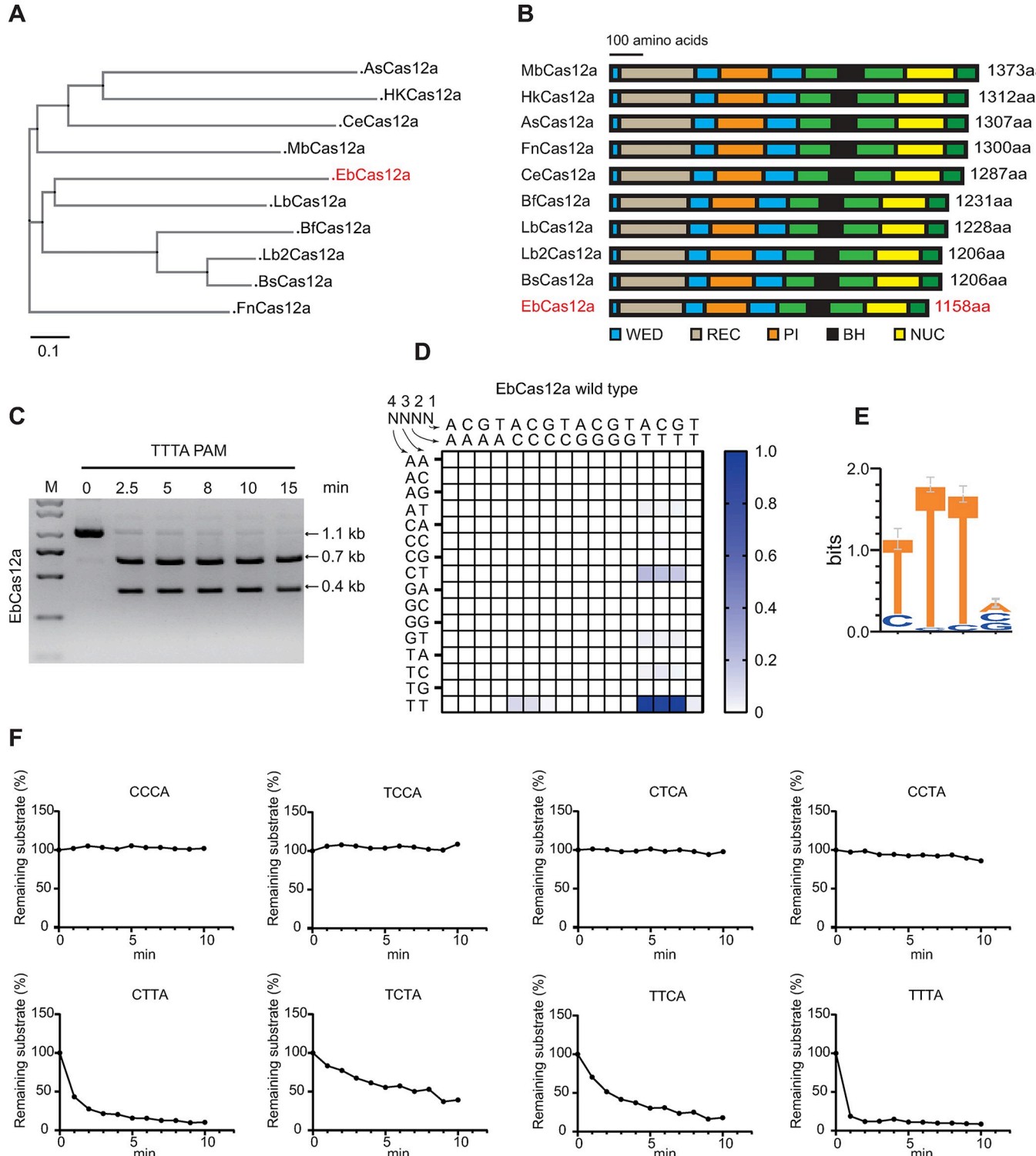

**Fig 1. DNA cleavage of EbCas12a in vitro. (A)** A phylogenetic tree generated by Phylo.io based on an alignment of Cas12a orthologs of the indicated species. **(B)** Schematic diagram of the domain organization of the indicated Cas12a orthologs. WED: crRNA binding and processing domains (blue). REC: DNA binding domains. PI: PAM-interacting domain (brown). RuvC: DNA cleavage domains. BH: bridge helix (black). Nuc: DNA processing domain (yellow). Number at right indicates the number of amino acids of each Cas12a ortholog. **(C)** DNA cleavage activity of EbCas12a towards TTTA PAM substrate in vitro. **(D)** Normalized cleavage rates for all 4-base PAMs for EbCas12a. The intensity of color represented the activity of EbCas12a nuclease. **(E)** Web logo for the EbCas12a PAM. **(F)** Quantification of time course in vitro cleavage reactions of EbCas12a on linearized dsDNA substrates. Those were conducted at 11 time points, respectively. Curves were fit using the one-phase exponential decay equation. The data underlying this figure can be found in S1 Data.

also targeted during the PAM depletion assay (**Fig 1D and 1E**). Therefore, the preference of EbCas12a for C-containing PAMs was tested using different linear dsDNA substrates. The in vitro DNA cleavage assay showed that CTTV, TCTV, and TTCV PAMs could also be recognized (**Figs 1F and S1B**).

## EbCas12a exhibits extensive gene-editing activities in mammalian cells

Unlike the sgRNA for Cas9, the guide RNA for Cas12a is only a single crRNA, making it easier for genome editing applications [27]. The crRNA for EbCas12a consists of a spacer and a direct repeat sequence. Considering that the structure of the crRNA might affect the efficiency of Cas12a genome editing, we investigated the spacer length requirements and the optimal direct repeat for EbCas12a. Firstly, we constructed a series of crRNAs with different lengths of spacer sequence (13, 15, 17, 19, 21, 23, 25, 27, 29, and 31 nt) to target the same site in *EGFP*. The EGFP disruption assay showed that crRNAs with spacer lengths ranging from 21 to 25 nt could efficiently mediate genome editing (**Figs 2A and S2**). Furthermore, 9 crRNAs with different direct repeats but the same spacer sequence were tested in the EGFP disruption assay. The most efficient EGFP disruption was observed by EbCas12a with its cognate crRNA and Lb2 crRNA (**Figs 2A and S2**). Subsequently, we explored the cleavage activity of endogenous genes in human cells. From the T7E1 assay and target deep sequence at the *PRKCH* target site, the result showed that EbCas12a could efficiently perform the gene editing for the endogenous gene (**Fig 2C and 2D**). Furthermore, we selected 13 endogenous loci and verified their targeting by EbCas12a using the T7E1 assay. This analysis demonstrated that EbCas12a generated indels at all 13 sites in HEK293T cells, with mutation rates ranging from 6% to 27% (**Fig 2E and 2F**). Collectively, our findings suggest that EbCas12a exhibits DNA-cleaving activity in human cells.

## EbCas12a-D141R has higher editing efficiency than wild-type EbCas12a

Our analysis indicated that while EbCas12a possesses some ability to introduce indels in endogenous genes, the activity was relatively low (usually less than 20%). We attempt to find the more efficient version of EbCas12a. By aligning its amino acid sequence with those of other more effective Cas12a orthologs, the key mutation (D141R) was selected and the novel variant (enEbCas12a) was engineered [28] (**Fig 3A**).

Through EGFP disruption assay [29,30], it was demonstrated that enEbCas12a (EbCas12a with the D141R mutation) displayed increased nuclease activity relative to the unmodified EbCas12a (**Fig 3B**). Furthermore, we analyzed the editing efficiency of enEbCas12a and wild-type EbCas12a at 7 additional genomic loci using both T7E1 assay and targeted deep sequencing. The resulting data illustrated that enEbCas12a surpassed wild-type EbCas12a across all examined targets (**Figs 3C, 3D, and S3–S6**). Next, we compared the activities of wild-type EbCas12a, enEbCas12a, AsCas12a, enAsCas12a-HF1, LbCas12a, and Lb2Cas12a across several endogenous sites in HEK293T cells. The T7E1 assay was used to rapidly analyze the efficiency for each Cas12a. enEbCas12a showed higher efficiency than wild-type EbCas12a, comparable to AsCas12a and LbCas12a (**Figs 3E, S7, and S8**). The enEbCas12a could also generate indels in Hepa1-6 cells with high efficiency (**S9 Fig**). Overall, these results demonstrate that enEbCas12a has great potential as a powerful tool for genome editing due to its improved gene editing capabilities.

## Genome editing by enEbCas12a shows high-fidelity

Potential off-target effects can be serious hurdles for the CRISPR/Cas system and may limit its applications in gene therapy. Previous reports have demonstrated that AsCas12a and

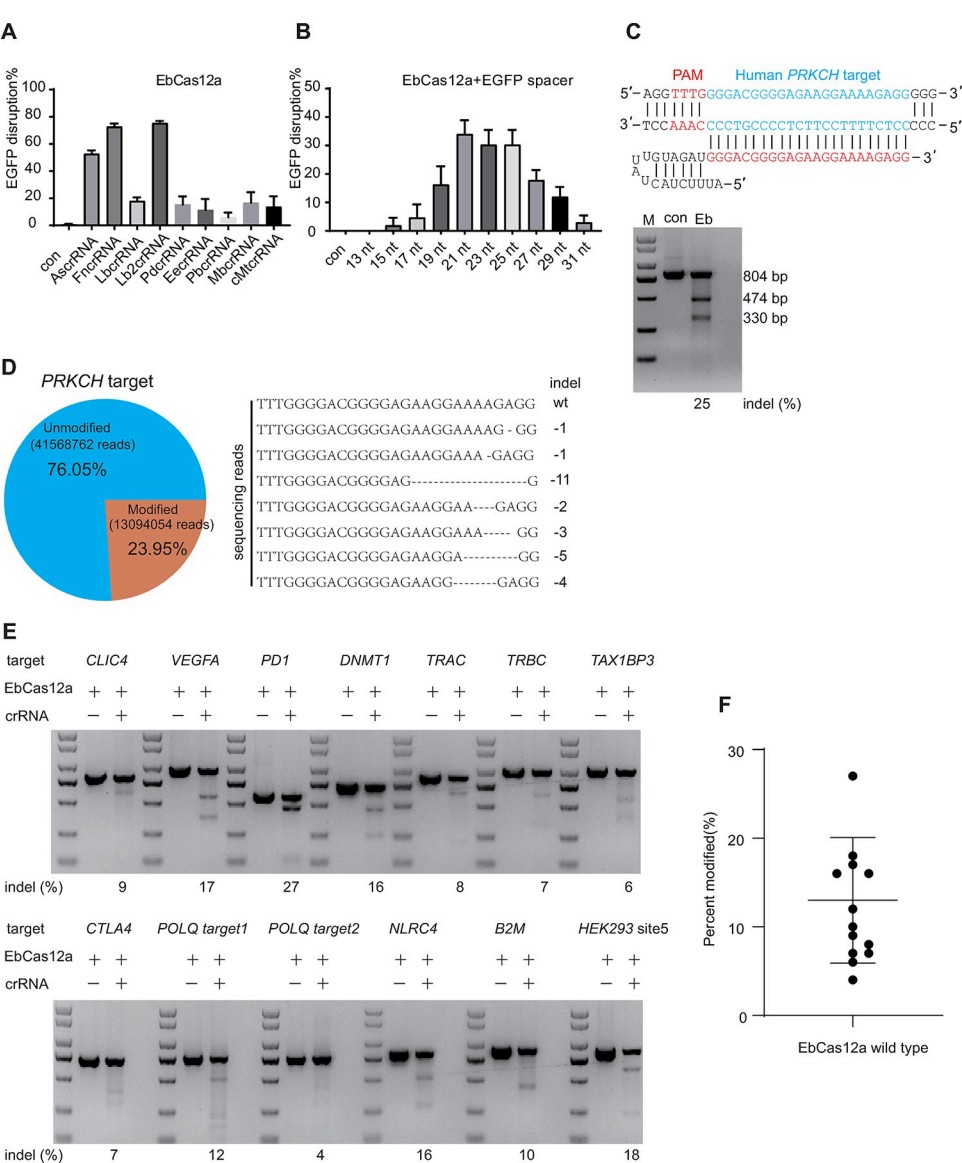

**Fig 2. EbCas12a mediates gene editing in vivo.** **(A)** The suitability of EbCas12a with crRNAs containing different loop regions from Cas12a orthologs. **(B)** EGFP disruption mediated by EbCas12a and crRNAs bearing variable-length complementarity regions for the target site of EGFP in human cells. **(C)** Schematic showing the sequence of crRNA targeting the human *PRKCH* (Upper). T7E1 analysis of targeted indel frequencies induced by the EbCas12a (Lower). **(D)** Pie chart showing the deep sequencing read counts of *PRKCH* locus (Left). Sequencing reads show representative mutations of EbCas12a-mediated gene editing with *PRKCH* crRNA (Right). Dashes represent the DNA deletions. The number at the right side of each sequence is the length of indel (−, deletion). **(E)** T7E1 analysis of targeted indel frequencies induced by the EbCas12a with their own crRNA in 13 genomic loci. **(F)** Summaries of the activities of EbCas12a at TTTV PAMs from **E**. The data underlying this figure can be found in S1 Data. crRNA, CRISPR RNA.

LbCas12a have low rates of off-target activity in human cells. To determine if enEbCas12a displays low off-targeting during genome editing, we selected 7 endogenous gene targets and compared their performance with those of AsCas12a, LbCas12a, and enEbCas12a under the genome-wide unbiased identification using the GUIDE-seq method [31].

Overall, among the 7 loci targeted by enEbCas12a, no off-target site was detected for *B2M* and *CTLA4* loci, while 1 off-target site was detected for *NLRC4* and *CLIC4* loci, 2 off-target

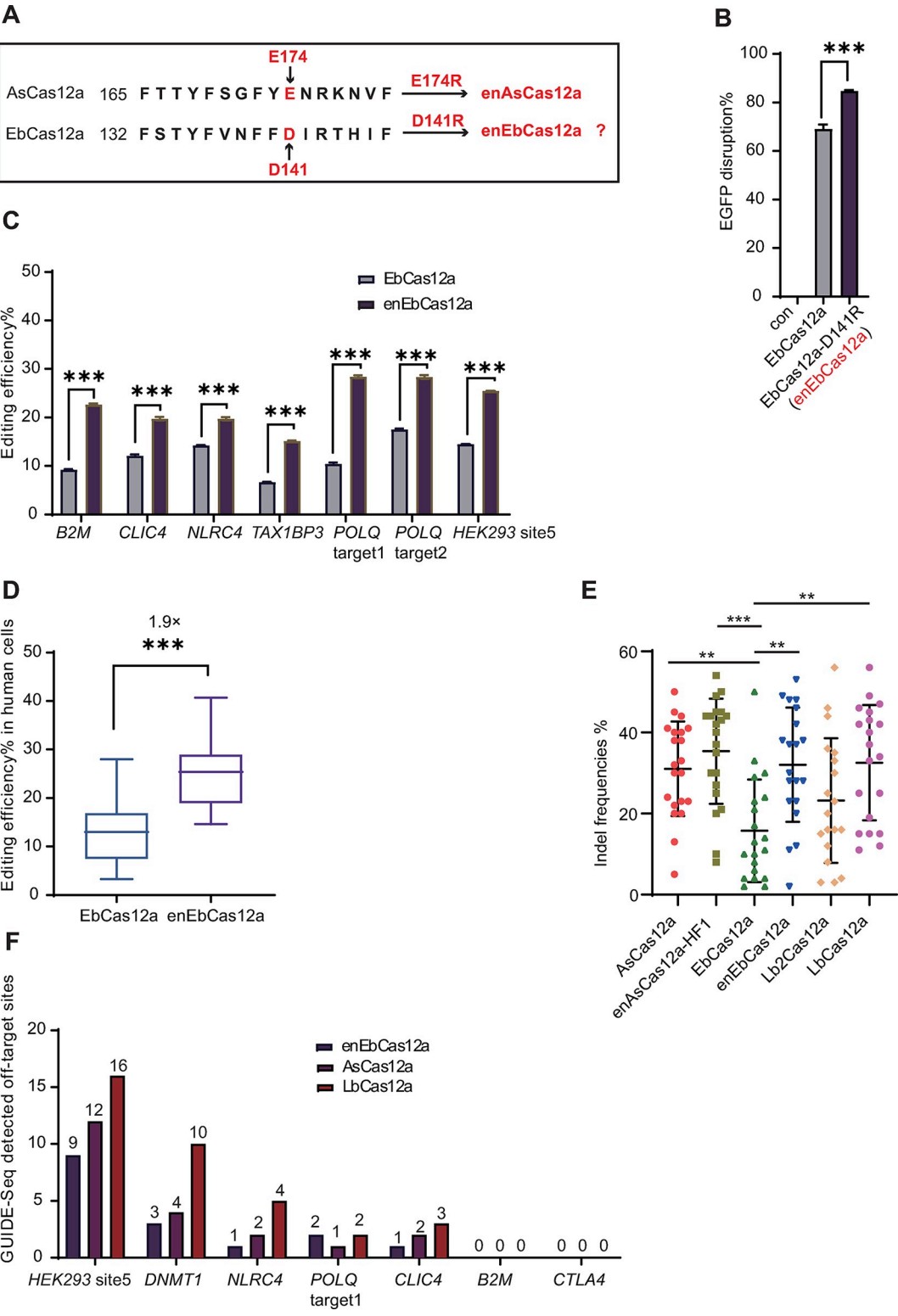

**Fig 3. Enhanced editing efficiency of EbCas12a variant. (A)** Sequence alignment of EbCas12a and enAsCas12a. Arrows indicate the E174 of LbCas12a and D141 of EbCas12a, respectively. **(B)** Efficiencies of EGFP disruption mediated by EbCas12a and enEbCas12a. Error bars represent SEM, *n* = 3. ***P < 0.001 (Mann–Whitney). **(C)** Efficiencies of enEbCas12a at TTTV PAMs at a diverse panel of target sites in HEK293T cells. For indel percentages, each column represents the mean of *n* = 3 transfected cell cultures. Indel frequencies were measured by deep sequencing. ***P < 0.001 (Mann–Whitney). **(D)**

Summaries of the activities of enEbCas12a at TTTV PAMs from **C**. (**E**) Comparison of enEbCas12a and Cas12a orthologs' gene-editing efficiencies. Indel frequencies analyzed by T7E1 assay. Gel images can be found in S8 Fig. (**F**) Histograms illustrating the number of GUIDE-seq detected off-target sites for enEbCas12a, AsCas12a, and LbCas12a. The data underlying this figure can be found in S1 Data.

sites for *POLQ1* locus, 3 off-target sites for *DNMT1* locus, and 12 off-target sites for HEK293 Site5 locus were detected. We simultaneously examined the off-target effects of AsCas12a and LbCas12a and similar or more off-target sites were detected for AsCas12a and LbCas12a (**Figs 3F and S10–S12**). In order to validate the off-target sites identified by GUIDE-seq, we used deep sequencing to confirm the occurrence of the indels (**S13 Fig**). This result is also consistent with the previously reported high specificity of Cas12a in gene editing [17,32]. Taken together, these results indicate that enEbCas12a combines both efficiency and fidelity in gene editing, making it an attractive candidate for therapeutic applications.

### In vivo genome editing using all-in-one AAV-enEbCas12a

The compact nature of the enEbCas12a system may enable it to be packaged within a single AAV vector for efficient delivery. To evaluate the efficacy of this approach, we engineered the AAV vector backbone to drive enEbCas12a expression with the EFS promoter and crRNA expression with the U6 promoter (**Fig 4A**). By comparing the titers of several AAV-Cas12a-crRNA viruses, we found that the AAV-enEbCas12a-crRNA virus particles were well packaged (**S14A–S14C Fig**). Firstly, we verified the activity by targeting *PD1* in 293T cells (**Fig 4B**). Then, we targeted *PCSK9* in mice using hepatotropic AAV serotype 9. The *PCSK9* has been previously indicated to be a good therapeutic target to reduce blood cholesterol levels [33,34]. Five target sites in *PCSK9* were screened in mouse hepa1-6 cells to obtain the optimal one (**Fig 4C**). Mice were divided into 2 independent cohorts and received either $2 \times 10^{11}$ genomic copies (GCs) of AAV-enEbCas12a-PCSK9-crRNA or phosphate-buffered saline (PBS) intravenously via the tail vein. The serum was collected for analysis at 0 and 30 days after injection, and all mice were euthanized 30 days after injection. One month after administration, we observed that the serum cholesterol level of AAV-enEbCas12a group decreased significantly, while PBS-treated mice maintained normal cholesterol level 30 days after injection (**Fig 4D**). In addition, indels were detected at the *PCSK9* locus in mouse liver by both the T7E1 assay and deep sequencing (**S14D and S14E Fig**). These data confirm that enEbCas12a can be used as an effective genome-editing system in vivo through a single AAV delivery.

### Discussion

In this study, we identified a compact Cas12a ortholog, EbCas12a, from the metagenome-assembled genome of a currently unclassified *Erysipelotrichia* with 1158aa, which can mediate genome editing in human cells. Through a reasonable introduction of a D141R mutation, we further developed enEbCas12a, which greatly increased its cleavage activity and have similar cleavage activities compared to AsCas12a and LbCas12a. GUIDE-seq data showed that enEbCas12a, like AsCas12a and LbCas12a, is also a highly accurate nuclease in gene editing. In addition, we optimized the components of the enEbCas12 system and successfully packaged enEbCas12a with its crRNA into an all-in-one AAV vector. Overall, our findings demonstrate the potential of enEbCas12a as a promising genome-editing tool and provide a novel example for Cas12a packed AAV in gene therapy.

Cas9 and Cas12a are the 2 most widely used gene-editing systems. Because of its low off-target effects, Cas12a is very promising for future applications in disease treatment [32,35]. Currently, packaging Cas protein into AAV for in vivo gene editing is one of the best methods for

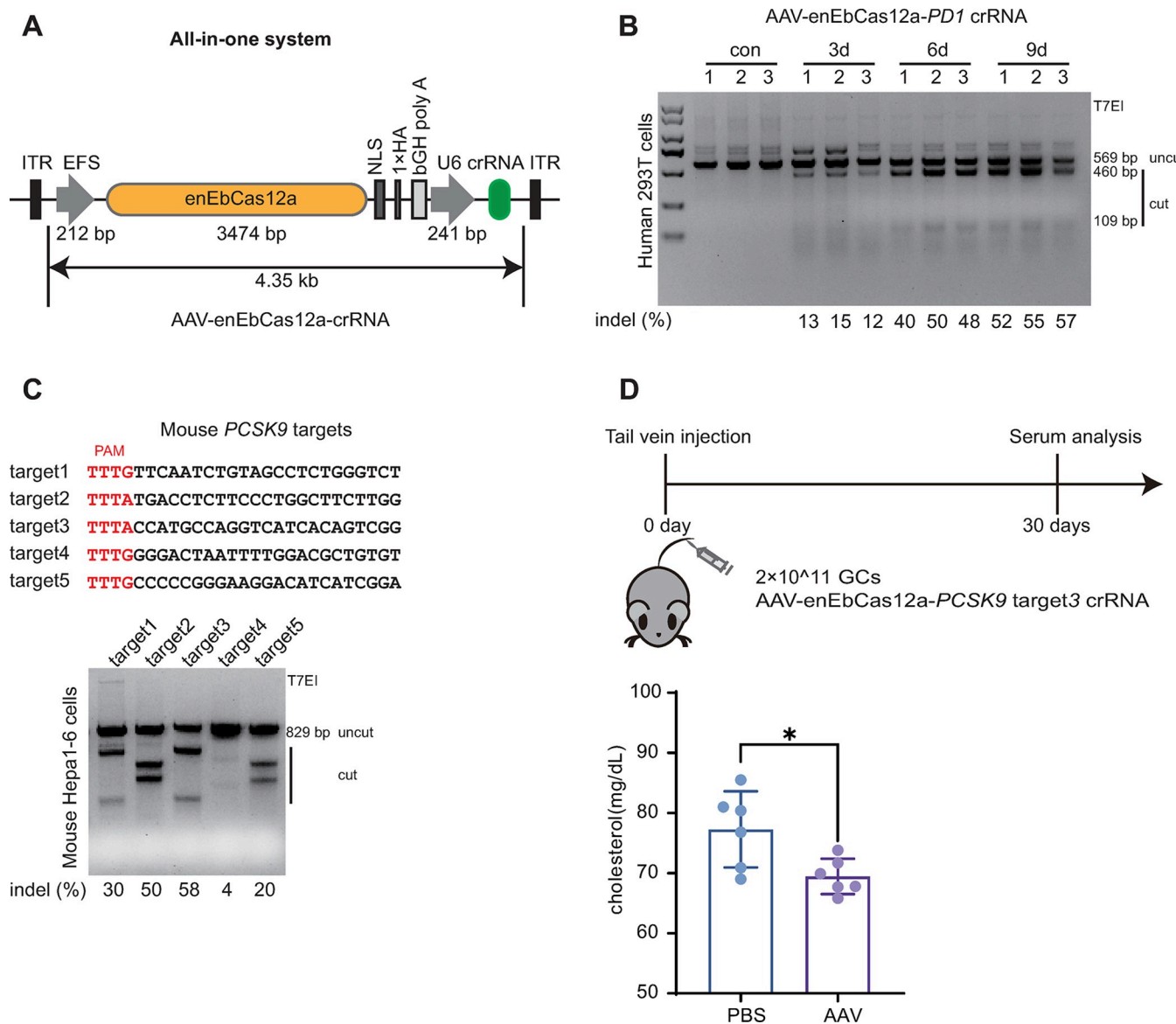

**Fig 4. All-in-one AAV delivery of enEbCas12a for in vivo genome editing. (A)** Single-vector AAV system. **(B)** Evaluation of All-in-one system. Indels at *PD1* target in human 293T cells following transduction of AAV at $2 \times 10^{11}$ total genome copies. Indel frequencies analyzed by T7E1 assay. Three replicates for each timeline. **(C)** Selection of *PCSK9* target sites. Sequences of *PCSK9* targets (Upper). Frequencies of indels at *Pcsk9* targets (Lower). Indel frequencies analyzed by T7E1 assay. **(D)** Experimental timeline of AAV-enEbCas12a-*PCSK9* target3 crRNA tail-vein injections. Analyses serum cholesterol levels in mice injected with AAV-enEbCas12a-*PCSK9* target3 crRNA (*n* = 6 animals) compared to the PBS controls (*n* = 6 animals). The data underlying this figure can be found in S1 Data. AAV, adeno-associated virus; crRNA, CRISPR RNA; PBS, phosphate-buffered saline.

clinical applications [22]. To overcome the large size of the Cas protein, researchers even used spilt-Cas protein and packaged it separately into AAV for in vivo editing [36]. Meanwhile several small Cas9 nucleases have been placed into a single AAV-Cas9-sgRNA system for genome editing [13,24,25]. However, Cas12a nucleases fail to fit inside AAV along with their crRNA because of their larger sizes and the small packaging capacity of AAV. The Compact EbCas12a is smaller than all the commonly used Cas12a to date. Wild-type EbCas12a could cleave endogenous genes but have low efficacy, which may limit its application. We improved the editing efficiency of EbCas12a by introducing a D141R mutation, which is located in the PAM

recognizing region and may increase DNA-binding affinity by raising interactions between the protein and the PAM sequence [28,37]. Genome editing of multiple endogenous gene sites showed that enEbCas12a has comparable editing activities to AsCas12a and LbCas12a. By reasonably engineering the enEbCas12a system, we optimized the total size of enEbCas12a and its crRNA, which is below 4.7 kb in length, and they were successfully encapsulated in an all-in-one AAV vector for in vivo genome editing.

Although a number of small Cas12f homologs [38,39], such as AsCas12f1 [39,40], Un1Cas12f1 [41,42], CWCas12f1 [43], OsCas12f1 [44], and RhCas12f1 [44], and a number of small Cas9 homologs [45], such as St1Cas9 [45–48], Nm1Cas9 [45,49–51], SaCas9 [46,52–55], CjCas9 [24,56], and Nm2Cas9 [57], SauriCas9 [58], sRGN3.1 [59], and SlugCas9 [59,60] have been identified and engineered to modify genomic DNA, Cas12a nucleases have been previously characterized as a generally low off-targeting family and have a unique ribonuclease activity to process their own crRNAs from longer precursors [27]. In addition, a potentially important difference in biochemical behavior between Cas12a nucleases and Cas9 nucleases is the ability of Cas12a to cleave nonspecifically single-stranded DNA (ssDNA) (cis-activity) following crRNA-guided target DNA binding and cleavage (trans-activity) [61–63]. Therefore, the distinct features may allow enEbCas12a to exhibit high fidelity and edit multiple genes simultaneously and potentially facilitate diverse virus detection.

Taken together, we identified the compact EbCas12a with 1158aa and obtained the enEbCas12a that can induce efficient genome editing in cells. Moreover, the compact size of enEbCas12a enables it to be delivered in vivo by an all-in-one AAV vector for genome editing. The enEbCas12a greatly expanded the toolbox for gene therapy and provided a promising platform for AAV-mediated gene therapy with Cas12a-based systems.

## Materials and methods

### Animals

C57BL/6J female mice were purchased from Beijing Vital River Laboratory Animal Technology Co. (Beijing, China). All mice were maintained in the specific pathogen-free animal facility at Wuhan University. All experimental protocols for the animal studies were approved by the Ethics Committee for Animal Experimentation of Wuhan University and were in accordance with the institutional guidelines (protocol code WDSKY0201603-2 approved October, 2020).

### Plasmid construction

EbCas12a was obtained from the metagenome-assembled genome of a currently unclassified *Erysipelotrichia* [26] (GenBank accession number: SAAR01000186; Taxonomy ID: 237 2184014, Genome assembly: ASM991633v1). The codon-optimized EbCas12a gene (was synthesized by Genecreate (Wuhan), then cloned into pET28a(+) (NcoI, XhoI) for protein expression and cloned into pcDNA3.1(+) (EcoI, HindIII) for expression in cells. The CRISPR templet was synthesized by Genecreate (Wuhan) and cloned into pCDFDuet-1(NdeI, XhoI). AsCas12a, LbCas12a, FnCas12a, and Lb2Cas12a plasmids were purchased from Addgene (#69982 and #69988). Plasmids for Cas12a crRNA were generated by synthesizing spacer sequence and cloned into pU6-Fn-crRNA plasmids containing a U6 promoter. EbCas12a variant was generated via PCR-amplified DNA fragment containing the point mutation and then inserted into wild-type EbCas12a plasmid instead of the original sequence. All sequences used in this study can be found in S1 Table.

## Protein expression and purification

pET28a-EbCas12a and pCDFDuet-CRISPR array bacterial expression vector were co-expressed in *E. coli* Rosetta [64]. Transformed *E. coli* were cultured at 37˚C, 200 rpm/min in Luria–Bertani (LB) medium until their optical density reached 0.6. Subsequently, cultures were induced with 0.3 mM isopropyl b-D-thiogalactoside (IPTG) at 16˚C, 150 rpm/min for 16 h. Bacterial cells were harvested by centrifuging at 4˚C, 4,000 g for 10 min, and then used buffer A (10 mM Tris-HCl (pH 8.0), 5 mM $MgCl_2$, 200 mM NaCl, 5 mM imidazole, 0.1% Triton-100) to resuspend them. After disrupting bacterial cells by the hydraulic breaker, the supernatant of cell lysate was collected by centrifuging at 4˚C, 10,000 g for 15 min, and then incubated with the Ni-NTA resin at 4˚C for 30 min. EbCas12a-crRNA RNP was washed with buffer B (10 mM Tris-HCl (pH 8.0), 5 mM $MgCl_2$, 200 mM NaCl) and eluted with buffer C (10 mM Tris-HCl (pH 8.0), 5 mM $MgCl_2$, 200 mM NaCl, 200 mM imidazole), and then exchanged with buffer B, stored at −80˚C.

## Cas12a DNA cleavage assay in vitro

DNA substrates were amplified by PCR from the pTriex-EGFP plasmid. In vitro cleavage was performed in buffer (20 mM HEPES-NaOH (pH 7.5), 100 mM KCl, 2 mM $MgCl_2$, 1 mM DTT, and 5% glycerol) containing 300 ng DNA substrate and 150 nM Cas12a-crRNA RNP at 37˚C. The reaction was terminated by the addition of Protease K at 58˚C for 30 min and resolved on 2% agarose gels.

## PAM characterization assay

The PAM library was created by an equal amount of mixed DNA fragments containing a randomized 5′-NNNN-3′ in PAM. The EbCas12a-crRNA complex was incubated at 37˚C for 2.5, 5, 8, 10, and 15 min with 100 ng PAM library DNA in reaction buffer (20 mM HEPES-NaOH (pH 7.5), 100 mM KCl, 2 mM $MgCl_2$, 1 mM DTT, and 5% glycerol). The reaction was terminated at 95˚C for 10 min. After purifying the reaction product, 1 μl product was amplified by PCR using Illumina primers.

## Cell culture and transfection

HEK293T cells and hepa1-6 cells were cultured in DMEM with 10% FBS and 1% penicillin-streptomycin at 37˚C with 5% $CO_2$. The 24-well plate cells were transfected with 500 ng Cas12a plasmid, 250 ng crRNA plasmid using 2 μl Hieff Trans Liposomal Transfection Reagent (CAT:40802ES03, Yeasen, Shanghai) for indel analysis. The 96-well plate cells were transfected with 100 ng Cas12a plasmid, 50 ng crRNA plasmid using 2 μl Hieff Trans Liposomal Transfection Reagent for EGFP disruption assay analysis. All cells were transfected at 70% to 80% confluency.

## T7E1 assay

Two days after transfection, cells were harvested to extract genomic DNA using Animal Tissue Direct PCR Kit (cat. no.: 10180ES70, Yeasen, Shanghai). Target sites were amplified from extracted genomic DNA by PCR and purified using SanPrep Column PCR Product Purification Kits (Sangon Biotech). A total of 250 ng of purified PCR products were denatured, reannealed, and then incubated with T7E1 enzyme (cat. no.: EN303-02, Vazyme, Nanjing) at 37˚C for 20 min. The reaction products were resolved on 2% agarose gel.

### Target deep sequencing assay

As described above, target sites were amplified from extracted genomic DNA using Illumina primers. PCR products with read length of about 250 bp were sent to the company for deep sequencing.

### Virus production and titration

For virus production, HEK293T cells were grown in 150 mm plates in Dulbecco's modified Eagle's medium (DMEM) supplemented with 10% FBS at 37°C with 5% CO2 (30 150 mm plates per Cas12a). For each 150 mm plate transfection, 7 μg of pAAV9 serotype packaging plasmid, 20 μg of pDF6 helper plasmid, and 7 μg of AAV2 plasmid carrying the construct of interest were added to 1 ml of serum-free DMEM. A total of 68 μl of Hieff Trans Liposomal Transfection Reagent was added to the mixture and incubated for 20 min at room temperature, then added to each 150 mm plate. After 8 h of incubation, 25 ml of warm maintenance medium was used to replace the old growth medium. Cells were harvested by scraping and pelleted by centrifugation 60 h after transfection. The AAV2/9 viral particles were then purified from the pellet according to a previously described procedure [65]. We used the plasmid of each AAV-Cas12a-crRNA for the real-time PCR standards, the concentration of which was measured by absorbance at 260 nm. To calculate the copy number, we used the molecular weight of the plasmid. We made serial dilutions of the standards in the range of $10^1$–$10^{-6}$ ng per 1 μl using ddH2O. For the titration of AAV2/9 viral particles, we performed a real-time quantitative PCR-based method as described previously [66].

### Animal injection and processing

AAV vial particles were administered intravenously to 5- to 6-week-old male C57/BL6 mice via lateral tail vein injection at a dose of $2 \times 10^{11}$ genomic copies per mouse. Prior to injection, the AAV dose was adjusted to 150 μl or 200 μl with sterile PBS (Gibco) at pH 7.4. To track the serum levels of total cholesterol, we performed orbital blood sampling on mice at various time points. After allowing the blood to clot at room temperature, the serum was separated by centrifugation and stored at −20°C for subsequent analysis. Serum total cholesterol measured using Infinity Cholesterol Reagent (Thermo Fisher) according to manufacturer's instructions, and $1 \times 1 \times 3$ mm$^3$ liver was cut into pieces and then lysed for indels analysis.

### GUIDE-seq

The 500 ng Cas12a plasmid, 250 ng crRNA plasmid, and 100 pmol of the double-stranded oligodeoxynucleotide (dsODN) GUIDE-seq tag were transfected into 24-well plate cells using 2.5 μl Hieff Trans Liposomal Transfection Reagent (CAT:40802ES03, Yeasen, Shanghai); 48 h after transfection, cells were harvested to extract genomic DNA using FastPure Cell/Tissue DNA Isolation Mini Kit DC102 (Vazyme, Nanjing). Approximately 1 μg of genomic DNA was sheared to a length of approximately 250 bp, added to a Y adapter, and amplified by 2 rounds of nested, anchored PCR. The last PCR products were GUIDE-seq libraries and sent to the company for sequencing using an Illumina sequencer.

### Statistics

Data were analyzed by Mann–Whitney tests via GraphPad Prism version 8.0. All data are expressed as mean ± SEM of at least 3 independent experiments. The probability values are reported using GraphPad style: not significant (ns), $P > 0.05$; *, $P < 0.05$; **, $P < 0.01$; ***, $P < 0.001$.

## Supporting information

**S1 Fig. Cleavage assay in vitro. (A)** Schematic of in vitro cleavage assay used to identify PAM sequence. **(B)** The preferences of EbCas12a toward different PAMs in vitro. The EbCas12a-crRNA complex (100 nM) was incubated at 37˚C for 8 min with 300 ng DNA substrates with the different PAMs (CCCN, CCTN, CTCN, CTTN, TCCN, TCTN, TTCN, TTTN), respectively. The data underlying this figure can be found in S1 Data.
(TIF)

**S2 Fig. crRNA Components. (A)** Schematic representation crRNA direct repeat structures. The difference among these 8 Cas12a family members is shown in shadow. **(B)** Schematic showing the sequence of *EGFP*-targeting crRNA. **(C)** Schematic showing variable length complementarity regions for the target site of *EGFP* in human cells.
(TIF)

**S3 Fig. enEbCas12a mediates gene editing in HEK293.** Gel image of Fig 3C. Assessment of gene-editing efficiencies with enEbCas12a. Activities assessed by T7E1 assay. Replicates represented transfected cell cultures times ($n$ = 3). The data underlying this figure can be found in S1 Data.
(TIF)

**S4 Fig. Measurements of gene-editing efficiencies of EbCas12a using deep sequencing.** Pie chart showing the deep sequencing read counts and indel frequencies of 7 endogenous gene targets.
(TIF)

**S5 Fig. Measurements of gene-editing efficiencies of enEbCas12a using deep sequencing.** Pie chart showing the deep sequencing read counts and indel frequencies of 7 endogenous gene targets.
(TIF)

**S6 Fig. Representative indels mediated by enEbCas12a.** Sequencing reads show representative mutations of enEbCas12a-mediated gene editing with 7 crRNAs. Dashes represent the DNA deletions. The number at the right side of each sequence is the length of indel (−, deletion).
(TIF)

**S7 Fig. Comparison of enEbCas12a and Cas12a orthologs' gene-editing efficiencies. (A)** Gel image of T7E1. Replicates represented transfected cell cultures times ($n$ = 3). **(B)** Indel frequencies analyzed by T7E1 assay. Summaries of the activities of Cas12a at TTTV PAMs from **A**. The data underlying this figure can be found in S1 Data.
(TIF)

**S8 Fig. Evaluation of the activity of AsCas12a, enAsCas12aHF1, EbCas12a, enEbCas12a, Lb2Cas12a, LbCas12a.** Activity of Cas12a variants on targets with candidate PAM sequences assessed by T7 endonuclease I assay. Average activity of Cas12a variants on targets are shown in Fig 3E.
(TIF)

**S9 Fig. enEbCas12a mediates gene editing in mouse Hepa1-6 cells. (A)** Indel mutations introduced at endogenous gene targets by enEbCas12a at TTTV PAMs. Indel frequencies of 14 gene targets were measured by T7E1 assay. **(B)** Summaries of the activities of enEbCas12a at TTTV PAMs from **A**. The data underlying this figure can be found in S1 Data.
(TIF)

**S10 Fig. Specificity analysis of matched AsCas12a targets.** GUIDE-seq analysis of detected off-targets for AsCas12a in Fig 3F. Mismatched positions are highlighted in color, and GUIDE-seq read counts are shown to the right of the on- or off-target sequences.
(TIF)

**S11 Fig. Specificity analysis of matched LbCas12a targets.** GUIDE-seq analysis of detected off-targets for LbCas12a in Fig 3F. Mismatched positions are highlighted in color, and GUIDE-seq read counts are shown to the right of the on- or off-target sequences.
(TIF)

**S12 Fig. Specificity analysis of matched enEbCas12a targets.** GUIDE-seq analysis of detected off-targets for enEbCas12a in Fig 3F. Mismatched positions are highlighted in color, and GUIDE-seq read counts are shown to the right of the on- or off-target sequences.
(TIF)

**S13 Fig. Deep sequencing validation the off-target sites identified by GUIDE-seq. (A)** Evaluation of the activity of AsCas12a, LbCas12a, and enEbCas12a. Activity of Cas12a variants on targets assessed by T7 endonuclease I assay. **(B)** Average activity of Cas12a variants on targets; summary of on-target modifications from A. **(C)** Percent modification GUIDE-seq detected off-target sites with indel mutations for AsCas12a, LbCas12a, and enEbCas12a. The data underlying this figure can be found in S1 Data.
(TIF)

**S14 Fig. Evaluation of the All-in-One AAV-Cas12a-crRNA System. (A)** Single vector AAV system. **(B)** Standard curve showing the data points for the dilution series of the AAV-Cas12a-crRNA plasmid. **(C)** The titer of AAV-Cas12a-crRNA viral particles. **(D)** In vivo genome editing using AAV-enEbCas12a-PCSK9-crRNA. Indels analyzed by T7E1 assay ($n$ = 3 animals for time points). **(E)** Sequencing reads show representative mutations of AAV-enEbCas12a-PCSK9-crRNA-mediated gene editing in liver. Dashes represent the DNA deletions. The number at the right side of each sequence is the length of indel (−, deletion). The data underlying this figure can be found in S1 Data.
(TIF)

**S1 Table. List of sequences used in study.**
(DOCX)

**S1 Data. Individual numerical values of Figs 1–4, S7, S9, S13, and S14.**
(XLSX)

**S1 Raw Images. Extended data figures of uncropped gels. Abbreviations:** PAM, protospacer adjacent motif; CRISPR-Cas, Clustered Regularly-Interspaced Short Palindromic Repeats and CRISPR-Associated proteins; EbCas12a, a Cas12a ortholog from a bacterial species of the class *Erysipelotrichia*; AsCas12a, *Acidaminococcus sp*. Cas12a; LbCas12a, *Lachnospiraceae bacterium*. Cas12a; RNP, ribonucleoprotein; SpCas9, *Streptococcus pyogenes*. Cas9; SaCas9, *Staphylococcus aureus*. Cas9; EGFP, enhanced green fluorescence protein; GUIDE-seq, genome-wide unbiased identification of double-stranded breaks enabled by sequencing; enEbCas12a, enhanced EbCas12a variant; enAsHF1, enhanced *Acidaminococcus sp*. Cas12a variant; WT, wild type; GCs, genomic copies.
(PDF)

## Acknowledgments

We thank all the members of our laboratory for the fruitful discussions and support.

## Author Contributions

**Conceptualization:** Jin Zhou, Yan Wang, Peng Chen, Lei Yin.

**Data curation:** Hongjian Wang, Jun Lei, Yankang Wu.

**Formal analysis:** Hongjian Wang, Yankang Wu, Yongshun Chen, Peng Chen.

**Funding acquisition:** Jin Zhou, Peng Chen, Lei Yin.

**Methodology:** Hongjian Wang, Jin Zhou, Jun Lei, Huan Liu, Ziyan Pang, Mingkun Du, Zihao Zhou, Zaiqiao Sun.

**Supervision:** Yan Wang, Peng Chen, Lei Yin.

**Validation:** Guosheng Mo.

**Visualization:** Hongjian Wang.

**Writing – original draft:** Jin Zhou.

**Writing – review & editing:** Chonil Paek, Lei Yin.

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
