## [Editor Report · Decision Letter 0]

13 Aug 2023

Dear Dr Yin, 

Thank you for submitting your manuscript entitled "The compact high-fidelity EbCas12a enables all-in-one AAV delivery system" for consideration as a Research Article by PLOS Biology. Please accept my apologies for the delay in getting back last week as we consulted with an academic editor about your submission.

Your manuscript has now been evaluated by the PLOS Biology editorial staff, as well as by an academic editor with relevant expertise, and I am writing to let you know that we would like to send your submission out for external peer review.

IMPORTANT: After discussions within the editorial team, we would like to consider your manuscript as a Short Report. Upon resubmission (details below), I would be grateful if you could tick 'Short Report' as the article type. 

Before we can send your manuscript to reviewers, we need you to complete your submission by providing the metadata that is required for full assessment. To this end, please login to Editorial Manager where you will find the paper in the 'Submissions Needing Revisions' folder on your homepage. Please click 'Revise Submission' from the Action Links and complete all additional questions in the submission questionnaire.

Once your full submission is complete, your paper will undergo a series of checks in preparation for peer review. After your manuscript has passed the checks it will be sent out for review. To provide the metadata for your submission, please Login to Editorial Manager (https://www.editorialmanager.com/pbiology) within two working days, i.e. by Aug 15 2023 11:59PM.

Kind regards,

Richard

Richard Hodge, PhD

rhodge@plos.org

PLOS

---

## [Decision Letter · Decision Letter 1]

2 Oct 2023

Dear Dr Yin,

Thank you for your patience while your manuscript "The compact high-fidelity EbCas12a enables all-in-one AAV delivery system" was peer-reviewed at PLOS Biology. Please accept my sincere apologies for the delays that you experienced during the peer review process. Your manuscript has now been evaluated by the PLOS Biology editors, an Academic Editor with relevant expertise, and by two independent reviewers. 

In light of the reviews, which you will find at the end of this email, we would like to invite you to revise the work to thoroughly address the reviewers' reports.

As you will see, the reviewers are generally positive about the findings but Reviewer #1 notes that the overall strength of the biochemical characterization of enEbCas12a should be strengthened. Specifically, this includes analysing Cas12a variants to compare editing efficiency and using deep sequencing to more robustly characterize potential off-target effects. After discussions with the Academic Editor, we will not make addressing major comment #2 essential for the revision but we encourage you to include this data if possible. 

Given the extent of revision needed, we cannot make a decision about publication until we have seen the revised manuscript and your response to the reviewers' comments. Your revised manuscript is likely to be sent for further evaluation by all or a subset of the reviewers.

**IMPORTANT - SUBMITTING YOUR REVISION**

*Re-submission Checklist*

*Published Peer Review*

*PLOS Data Policy*

*Blot and Gel Data Policy*

Sincerely,

Richard

Richard Hodge, PhD

rhodge@plos.org

REVIEWS:

Reviewer #1: Wang et al. demonstrated genome editing using a compact Cas12a, EbCas12a. They determined the TTTV PAM sequences of EbCas12a and developed engineered version of EbCas12a, enEbCas12a. Next, the activity and specificity of enEbCas12a were compared with compared with AsCas12a and LbCas12a. Lastly, the authors investigated a single AAV system with EbCas12a and crRNA in vitro and in vivo. 

1. In Figure 3E, the activity of enEbCas12a was compared only with wildtype AsCas12a and LbCas12a at seven target sites. To establish the utility of enEbCas12a, it would be necessary to measure and compare editing efficiency at a sufficient number of target sites where statistical significance can be demonstrated. I strongly recommend including the latest Cas12a variants, such as enAsCas12a (PMID = 30742127) or Lb2Cas12a (PMID = 33738137), for comparison.

2. If possible, it would be beneficial to perform an analysis of the features of target sites where enEbCas12a exhibits higher efficiency compared to other Cas12a variants.

3. To comprehensively analyze off-target effects, it's crucial to ensure comparable on-target activities. In Figure 3F, we require a clear description of the on-target activities for AsCas12, LbCas12a, and enEbCas12a. It would be more accurate to measure indel frequencies at endogenous target sites rather than relying solely on Guide-seq read counts. Furthermore, to validate the identified off-target sites using Guide-seq, it's essential to confirm the presence of indels at these potential off-target sites through deep sequencing.

4. Although enEbCas12a is relatively small with a size of 1158 amino acids compared to the Cas12a family, there have been much smaller genome editing tools developed recently, such as Un1Cas12f (529 aa, PMID = 34475560) and AsCas12f (422 aa, PMID = 37400536). Among Cas9 variants, sRGN3.1 (1057 aa) and SlugCas9 (1054 aa) that showing robust activity (PMID = 37188955) also smaller than enEbCas12a . Therefore, an analysis and discussion of the advantages that enEbCas12a possesses compared to these smaller counterparts is needed.

Reviewer #2: The authors have cloned and harnessed an enhanced variant of EbCas12a and demonstrate its compatibility with AAV vectors in cultured cells and in the mouse liver. The data are encouraging as their Cas12a version alleviates previous concerns about the size of Cas12a that have hampered packaging in and delivery by AAV.

Overall, this is an interesting article and a useful advance, thus I only have a few minor comments:

1) Did the EbCas12a vector package well? What were the average titers as compared to other vectors in the authors' hands?

2) Which dose was actually delivered to the mice? It says 2e11 in the text but 5e11 in the figure, and I couldn't find additional information in the Methods.

3) The drop in cholesterol levels in the mice is not overly impressive albeit significant. It would be good if the authors could further support and illustrate vector efficacy by showing the results of T7E1 assays from the mouse livers and quantifying the extent of target editing. Does it correlate with the mild drop in cholesterol?

---

## [Decision Letter · Decision Letter 2]

13 Feb 2024

Dear Dr Yin,

Thank you for your patience while we considered your revised manuscript "The compact high-fidelity EbCas12a enables all-in-one AAV delivery system" for publication as a Short Report at PLOS Biology. Please accept my apologies for the delays that you have experienced during this round of the peer review process. This revised version of your manuscript has been evaluated by the PLOS Biology editors, the Academic Editor and the original reviewers.

Based on the reviews, I am pleased to say that we are likely to accept this manuscript for publication, provided you satisfactorily address the following data and other policy-related requests that I have provided below (A-G). In addition, we ask that you please improve the quality of the writing/language in the Abstract before publication, by enlisting the services of a professional editing services or the help of an English-speaking colleague: 

(A) We would like to suggest the following modification to the title: 

““Engineering of a compact, high-fidelity EbCas12a variant that can be packaged with its crRNA into an all-in-one AAV vector delivery system”

(B) In the animal ethics statement provided in the Methods section, please provide the specific approval number issued by the Institutional Animal Care and Use Committee of Wuhan University.

(C) You may be aware of the PLOS Data Policy, which requires that all data be made available without restriction: http://journals.plos.org/plosbiology/s/data-availability. For more information, please also see this editorial: http://dx.doi.org/10.1371/journal.pbio.1001797

-Supplementary files (e.g., excel). Please ensure that all data files are uploaded as 'Supporting Information' and are invariably referred to (in the manuscript, figure legends, and the Description field when uploading your files) using the following format verbatim: S1 Data, S2 Data, etc. Multiple panels of a single or even several figures can be included as multiple sheets in one excel file that is saved using exactly the following convention: S1_Data.xlsx (using an underscore).

-Deposition in a publicly available repository. Please also provide the accession code or a reviewer link so that we may view your data before publication. 

Figure 1E-F, 2A-B, 2F, 3B-F, 4D, S7B, S8B, S9B, S13B-C, S14B-C

(D) Thank you for depositing the GUIDE-seq in the GEO database (GSE236647), but I would be grateful if the deep sequencing data could also be included in this deposition.

(E) Please also ensure that each of the relevant figure legends in your manuscript include information on *WHERE THE UNDERLYING DATA CAN BE FOUND*, and ensure your supplemental data file/s has a legend.

(F) We require the original, uncropped and minimally adjusted images supporting all blot and gel results reported in the following Figures:

Figure 1C, 2C, 2E, 4B-C, S1B, S3, S7A, S8A, S9A, S13A, S14D

We will require these files before a manuscript can be accepted so please prepare and upload them now. Please carefully read our guidelines for how to prepare and upload this data: https://journals.plos.org/plosbiology/s/figures#loc-blot-and-gel-reporting-requirements

(G) Please ensure that your Data Statement in the submission system accurately describes where your data can be found and is in final format, as it will be published as written there. 

We expect to receive your revised manuscript within two weeks. 

*Published Peer Review History*

*Press*

Kind regards,

Richard

Richard Hodge, PhD

rhodge@plos.org

Reviewer remarks:

Reviewer #1: In the revised manuscript, the authors have conducted a comprehensive evaluation of the compact, high-fidelity EbCas12a nuclease through additional wet experiments and analysis. The AAV-enEbCas12a described in the study would improve the efficacy and specificity of gene therapy. Thus, I recommend considering this study for publication in PLOS Biology.

Reviewer #2: I thank the authors very much for fully addressing my comments.

---

## [Editor Report · Decision Letter 3]

3 Apr 2024

Dear Lei,

Thank you for your patience while we considered your revised manuscript "Engineering of a compact, high-fidelity EbCas12a variant that can be packaged with its crRNA into an all-in-one AAV vector delivery system" for publication as a Short Report at PLOS Biology. 

Following on from our previous correspondence, I am writing to ask that you please provide some additional reporting details about the EbCas12a sequence in the manuscript before we can move forward to publication. I have listed these requests below (A-C), but please do let me know if you have any further questions about this:

(A) Thank you for clarifying that the EbCas12a sequence has been obtained from a metagenome-assembled genome of a currently unclassified Erysipelotrichia isolate. In the Genbank accession, it appears that this sequence was originally obtained in an metagenomic study in which the authors sequenced wastewater from ponds of two small diary industries in Argentina (BioProject: PRJNA508305). This dataset was initially deposited in 2020 and then eventually published in the PeerJ in 2022 (Irazoqui et al, 2022, PeerJ, PMID 35310160). At this time, we ask that you please and cite and refer to/contextualize this prior work in the PeerJ that initially provided the metagenomic sequences in the manuscript. 

(B) Pleaser ensure you refer to the EbCas12a as being obtained from the metagenome-assembled genome of a currently unclassified Erysipelotrichia in the manuscript text. This will need to be changed throughout the manuscript text, figure legends, abstract etc where the text currently refers to "the Erysipelotrichia bacterium bacteria" or "the Erysipelotrichia bacterium". Specifically, we would suggest that you state that the Cas12a was obtained from the metagenome-assembled genome of a currently unclassified Erysipelotrichia (eg the first time in the paper and then in the Methods section), and then when it is subsequently referred to you can write "a Cas12a ortholog from a bacterial species of the class Erysipelotrichia" or similar. 

(C) Please include the Genbank and Taxonomy ID accession numbers in the Data Availability Statement in the online submission form.

We expect to receive your revised manuscript within two weeks. 

*Published Peer Review History*

*Press*

Kind regards,

Richard

Richard Hodge, PhD

rhodge@plos.org

PLOS

---

## [Editor Report · Decision Letter 4]

9 Apr 2024

Dear Lei,

Thank you for the submission of your revised Short Report "Engineering of a compact, high-fidelity EbCas12a variant that can be packaged with its crRNA into an all-in-one AAV vector delivery system" for publication in PLOS Biology. On behalf of my colleagues and the Academic Editor, Bon-Kyoung Koo, I am pleased to say that we can in principle accept your manuscript for publication, provided you address any remaining formatting and reporting issues. These will be detailed in an email you should receive within 2-3 business days from our colleagues in the journal operations team; no action is required from you until then. Please note that we will not be able to formally accept your manuscript and schedule it for publication until you have completed any requested changes.

PRESS

Best wishes, 

Richard

Richard Hodge, PhD

rhodge@plos.org

PLOS
